# Longitudinal trajectories of blood lipid levels in an ageing population sample of Russian Western-Siberian urban population

**Jaroslav A. Hubacek**[1,2]*, **Yuri Nikitin**[3], **Yulia Ragino**[3], **Ekaterina Stakhneva**[3], **Hynek Pikhart**[4], **Anne Peasey**[4], **Michael V. Holmes**[5], **Denes Stefler**[4], **Andrey Ryabikov**[3], **Eugeny Verevkin**[3], **Martin Bobak**[4], **Sofia Malyutina**[3]

**1** Experimental Medicine Centre, Institute for Clinical and Experimental Medicine, Prague, Czech Republic, **2** 3[rd] Department on Internal Medicine, 1[st] Faculty of Medicine, Charles University, Prague, Czech Republic, **3** Research Institute of Internal and Preventive Medicine–Branch of Institute of Cytology and Genetics, Siberian Branch of Russian Academy of Sciences, Novosibirsk, Russia, **4** Department of Epidemiology and Public Health, University College London, London, United Kingdom, **5** Nuffield Department of Population Health, University of Oxford, Oxford, United Kingdom

* jahb@ikem.cz

**Data Availability Statement:** According to ethical restrictions as established in ICF and Approval protocol from Ethical Committee, we could not

## Abstract

This study investigated 12-year blood lipid trajectories and whether these trajectories are modified by smoking and lipid lowering treatment in older Russians. To do so, we analysed data on 9,218 Russian West-Siberian Caucasians aged 45–69 years at baseline participating in the international HAPIEE cohort study. Mixed-effect multilevel models were used to estimate individual level lipid trajectories across the baseline and two follow-up examinations (16,445 separate measurements over 12 years). In all age groups, we observed a reduction in serum total cholesterol (TC), LDL-C and non-HDL-C over time even after adjusting for sex, statin treatment, hypertension, diabetes, social factors and mortality (P<0.01). In contrast, serum triglyceride (TG) values increased over time in younger age groups, reached a plateau and decreased in older age groups (> 60 years at baseline). In smokers, TC, LDL-C, non-HDL-C and TG decreased less markedly than in non-smokers, while HDL-C decreased more rapidly while the LDL-C/HDL-C ratio increased. In subjects treated with lipid-lowering drugs, TC, LDL-C and non-HDL-C decreased more markedly and HDL-C less markedly than in untreated subjects while TG and LDL-C/HDL-C remained stable or increased in treatment naïve subjects. We conclude, that in this ageing population we observed marked changes in blood lipids over a 12 year follow up, with decreasing trajectories of TC, LDL-C and non-HDL-C and mixed trajectories of TG. The findings suggest that monitoring of age-related trajectories in blood lipids may improve prediction of CVD risk beyond single measurements.

## Introduction

Plasma lipid values are important modifiable risk factors of cardiovascular diseases [1]. According to the long-term used approach in preventive cardiology plasma total cholesterol

publicly share the individual-level data, even when they are anonymized. However, the data access is permanently available for the data manager of the HAPIEE project (UCL) and for the data manager of the NIITPM (Novosibirsk) and not solely for the named authors. Data are also available for UCL/ HAPIEE or NIITPM researchers by request and on condition of approval by the Steering Committee. Finally, by request from a third party, the project could provide tabulated data or, in exclusive cases, a restricted dataset for specific analysis and upon approval of the Steering Committee. Researchers may contact: the head of Ethical Committee at Novosibirsk University Professor Nikolaev K. Yu.: nikolaevky@yandex.ru and Dr. Wentien Lu, the data manager for the entire HAPIEE project - wentian.lu.14@ucl.ac.uk.

**Funding:** This study was supported by the Russian Scientific Foundation (20-15-00371,14-45-00030) and the Russian Academy of Science, State target (-17-117112850280-2); the HAPIEE study was funded by the Welcome Trust (WT064947, WT081081), the US National Institute of Aging (1RO1AG23522) and the MacArthur Foundation Initiative on Social Upheaval and Health; JAH is supported by the Ministry of Health of the Czech Republic under the Conceptual Development of Research Organisations project (Institute for Clinical and Experimental Medicine – IKEM, IN 00023001). MVH works in a unit that receives funding from the UK Medical Research Council and is supported by a British Heart Foundation Intermediate Clinical Research Fellowship (FS/18/ 23/33512) and the National Institute for Health Research Oxford Biomedical Research Centre. Sponsors play no role in the study design, data collection and analysis, decision to publish, or preparation of the manuscript.

**Competing interests:** The authors have declared that no competing interests exist.

(TC) over 5.0 mmol/L, plasma low-density cholesterol (LDL-C) over 3.0 mmol/l, plasma triglycerides (TG) over 1.7 mmol/L and plasma high-density cholesterol (HDL-C) below 1.0 mmol/L in males and below 1.2 mmol/L in females were recommended as optimal values for predicting increased risk of atherosclerosis development and subsequent cardiovascular disease. However, these values need to be modified according to further risk categories based on sex, smoking status, age and blood pressure [2]. The recent ESC/EAS guidelines on dyslipidaemia, 2019 defines the goal value of LDL-C in patients at very high, high, moderate, and low CVD risk as below 1.4 mmol/l, 1.8 mmol/l, 2.6 mmol/l, and 3.0 mmol/l, respectively [3].

Levels of plasma lipids are influenced both by genetic and environmental factors. For example, triglycerides seem to be subject to greater genetic influence than total cholesterol, with physical activity exerting more of an effect on triglycerides than dietary habits or smoking status [4]. To complicate matters further, all of these factors interact with each other [5].

Ageing has a considerable impact on lifestyle and quality of life; this is a particularly evident in Russia where life expectancy has increased dramatically over recent decades [6]. The side effect of this positive trend is the general increase in the prevalence of impaired health status in terms of non-fatal health outcomes. Despite this dramatic improvement in life expectancy in the Russian population, only scant information is available on the prevalence of dyslipidaemia, and reliable longitudinal analyses of individual-level changes in plasma lipids have yet to be performed.

Relatively few publications have explored long-term trajectories of plasma lipids [7–9], focusing primarily on Western European populations. Intriguingly, one recent study, albeit focusing on another ethnic group, suggested that different types of lipid trajectories in young and middle-aged subjects are associated with different risks of incident cardiovascular disease [10]. Consistent with this finding, differential risks of CVD were recently reported between clusters of individuals with different underlying lipid trajectories during 35-year follow-up in Framingham Offspring cohort [11].

The overall aim of our study was to track changes in blood lipid concentrations (mainly cholesterol in different fractions and triglycerides) associated with age in a Russian West-Siberian Caucasian (ethnic Russian) population. The cohort, based in Novosibirsk, the third-largest city in Russia, conducted repeated lipid examinations over a 12-year period. The secondary aim of the study was to determine whether and to what degree these changes would be influenced by smoking and lipid lowering treatment.

## Materials and methods

### Subjects

Participants in the Russian arm of the Health, Alcohol and Psychosocial Factors in Eastern Europe (HAPIEE) study were included in the analysis. The study sample comprised of men and women aged 45–69 at baseline living in two districts (Oktyabr'skiy and Kirovskiy) of the West-Siberian city of Novosibirsk, randomly samples from the electoral list stratified by 5-year age group and sex (response rate 61%). Details of the study protocol have been described elsewhere [12]; baseline characteristics of the subjects from the Russian part of the study have also been published previously [13, 14].

Anthropometric parameters, alcohol consumption, smoking status, history of dyslipidaemia and lipid-lowering treatment, history of cardiovascular and other chronic diseases, and socio-demographic parameters were assessed by trained nurses using a standardised protocol and questionnaire [12, 15]. Blood samples were drawn at fasting condition (at least 8 hours). Serum was stored at -80˚C and analyses for lipids were conducted within one month period

after samples collection. After the baseline examination in 2003–2005 (wave 1), participants were followed up by two subsequent waves in 2006–2008 (wave 2) and 2015–2017 (wave 3).

After excluding a minority of subjects aged 70 years or older at baseline (N = 111) from the analysis, the final study group consisted of 9,218 subjects (16,445 measurements/observations).

The study was conducted according to the guidelines of the Declaration of Helsinki, and was approved by the ethics committee of the Research Institute of Internal and Preventive Medicine—Branch of the Institute of Cytology and Genetics, Siberian Branch of Russian Academy of Sciences (protocol  1 from 14.mar.2002). Informed consent was obtained from all subjects involved in the study.

## Blood lipid analysis

Blood lipid concentrations across three waves were measured in Laboratory of Biochemistry, Institute of Internal and Preventive Medicine. Serum lipid values were analysed enzymatically using kits from Thermo Fisher Scientific (total cholesterol–Cat. No. 981812; HDL-cholesterol–Cat. No. 981823; triglycerides–Cat. No. 981301; LDL-cholesterol (LDL-C) direct measurement–Cat. No. 981656; HDL/LDL calibrator–Cat. No. 981657, and LIPOTROL control–Cat. No. 981653) on a KoneLab 30i autoanalyser (Thermo Fisher Scientific Inc., USA).

For subjects with serum TG values below 4.5 mmol/L, LDL-C values were calculated using the Friedewald formula [16]. For few subjects with TG values above 4.5 mmol/L, LDL-C values were measured using a direct enzymatic method. Non-HDL-C values were calculated using the equation: Non-HDL-C = TC–HDL-C.

Internal quality control was applied daily on a permanent basis by a biochemical laboratory, with external quality control routinely provided by a Moscow Federal standardisation centre on an annual basis. Laboratory was standardised for lipid measurements under the framework of the HAPIEE study, the Centres for Disease Control (Atlanta, USA) and the Institute of Clinical and Experimental Medicine (Prague, Czech Republic).

## Statistical analysis

In order to model trajectories of blood lipid values across three measurement waves over an average 12-year follow-up period, multilevel modelling (MLM) techniques were used. Specific measurement occasions at baseline, wave 2 and wave 3 were considered level 1 variables nested within study participants (level 2 variables). Both the intercept and slope were fitted as random effects, allowing individual differences in lipid concentrations at baseline and rate of change.

In order to distinguish between ageing and period/cohort effects, we separately modelled TC, LDL-C, HDL-C, TG as well as computed non-HDL-C and LDL-C/HDL-C trajectories over the 12-year follow-up period by 5-year birth cohorts. Estimates were unadjusted in model 1, adjusted for sex and lipid-lowering treatment (assessed in all three data collection waves) in model 2, and further adjusted for smoking (never-, ex-, current-smoker), BMI, alcohol consumption (non-drinkers, moderate drinkers, heavy drinkers: defined as >30 g/day for men and >15 g/day for women), education (primary, secondary, vocational, university), marital status (single, married/co-habiting), and CVD/all-cause mortality (identified by linkage with mortality-registry) in model 3. Slopes were calculated separately for male and female participants.

For sensitivity analysis, first, we recalculated trajectories only among subjects with available measurements of lipids across all three waves (N = 2,009). Second, we recalculated trajectories after exclusion of subjects who dropped out after baseline wave 1 and had no evidence of fatal outcome during follow-up period (N = 7,606).

All statistical analysis was carried out using MLwiN v3.01 software and accessed through Stata v13.1 by inputting the 'runmlwin' command [17]. P-values less than 0.05 were considered significant (calculated for trend).

## Results

### Descriptive characteristics of the sample

The general characteristics of the subjects evaluated across all three examinations are summarised in Table 1. Data on TC, LDL-C, HDL-C and TG were obtained from 9,218 participants at baseline, 3,442 subjects at wave 2 and 3,785 persons at wave 3, totalling 16,445 individual-level examinations. A total of 2,009 individuals were measured in all three waves (6,027 individual examinations). Waves 2 and 3 were conducted on average 3.2 and 12.3 years, respectively, after the baseline examination; the age range of participants increased from 45–69 years at baseline to 56–83 years at wave 3; however, the proportion of participants receiving lipid-lowering treatment (mostly statins) over time increased substantially between wave 1 and wave 3 from 23.8% to 47.1% among those aware of dyslipidemia.

### Overall lipid trajectories

Notable changes in serum lipids were observed over the follow-up period. In simple cross-sectional analysis by wave of measurement, we observed a moderate decrease in mean TC, LDL-C and HDL-C values between baseline and both follow-up phases for the entire population in both men and women. In contrast, mean TG values remained similar across all three examinations (Table 1).

Table 2 and Fig 1 show the unadjusted individual-level trajectories of blood lipids during follow up for all 5-year age groups. For in TC and LDL-C, the slopes of decline in concentrations were more pronounced in older cohorts, and this pattern was consistent with estimates adjusted for sex and lipid lowering treatment (Table 3) and with multivariable adjusted models (Table 4).

Changes in HDL-C over time were less pronounced compared to TC and LDL-C and the decline in HDL-C concentrations was similar in all 5-year groups (Tables 2–4; Fig 1), In contrast, TG values increased over the follow up in the younger cohort and decreased in the cohort of subjects over 60 years of age (Tables 2–4; Fig 1). For more details about the lipid trajectories in 5-years sub-group, see S1–S6 Figs.

As a sensitivity analysis, we repeated our assessment only in subjects whose serum lipids were obtained across all 3 waves ('complete cases', N = 2,009). The direction and slope of the trajectories virtually did not change (data are shown in S1 and S2 Tables).

Additionally, we repeated calculations in a subsample of 7,606 participants after exclusion of subjects who dropped out after baseline examination and did not die during follow-up. The direction and slope of the trajectories were similar to the patterns in the total sample (data are shown in S3 and S4 Tables).

### Effects of smoking status

Fig 2 shows the influence of smoking status on trajectories (ever- vs. never-smokers, adjusted for sex, lipid-lowering medication, and centred age at 58 years). At baseline and throughout the follow-up period, TC values remained comparable, exhibiting a similar decreasing slope in ever-smokers and never-smokers. Concentrations of LDL-C and non-HDL-C in smokers were lower at baseline and decreased less markedly than in non-smokers; in contrast, levels of HDL-C were higher among smokers at baseline but decreased more markedly. At baseline and throughout follow-up, TG was higher in smokers; trajectories were divergent, remaining

**Table 1. Descriptive characteristics of studied samples through 3 waves of examinations.**

| Characteristics | Wave 1 (2002–2005) n = 9,218 | | Wave 2 (2006–2008) n = 3,379 | | Wave 3 (2015–2017) n = 3,785 | |
|---|---|---|---|---|---|---|
| | men | women | men | women | men | Women |
| Observed, n | 4199 | 5019 | 1361 | 2018 | 1451 | 2334 |
| Time since W1, years | 0.0 | 0.0 | 2.95 | 3.33 | 12.2 | 12.4 |
| Age, years | 58.2 (6.94) | 57.9 (7.04) | 60.8 (6.91) | 61.0 (6.85) | 68.9 (6.89) | 69.4 (6.80) |
| TC, mmol/l | 6.01 (1.19) | 6.52 (1.30) | 5.55 (0.99) | 6.05 (1.09) | 5.18 (1.13) | 5.65 (1.18) |
| LDL-C, mmol/l | 3.84 (1.06) | 4.24 (1.17) | 3.52 (0.88) | 3.89 (0.99) | 3.29 (0.99) | 3.58 (1.08) |
| HDL-C, mmol/l | 1.50 (0.38) | 1.56 (0.35) | 1.42 (0.35) | 1.51 (0.33) | 1.24 (0.38) | 1.38 (0.38) |
| TG, mmol/l | 1.28 (0.97–1.74) | 1.38 (1.06–1.85) | 1.20 (0.92–1.58) | 1.27 (0.98–1.68) | 1.21 (0.86–1.74) | 1.30 (0.97–1.80) |
| SBP, mmHg | 144.2 (23.5) | 144.5 (26.1) | 145.2 (23.3) | 146.8 (25.3) | 146.6 (20.5) | 144.8 (21.6) |
| DBP, mmHg | 90.8 (13.4) | 90.5 (13.5) | 99.6 (13.0) | 90.3 (13.4) | 85.8 (11.8) | 82.2 (10.8) |
| BMI, kg/m$^2$ | 26.6 (4.44) | 30.2 (5.72) | 27.0 (4.45) | 30.3 (5.48) | 27.8 (4.61) | 30.5 (5.72) |
| LLT, n (%) * | 130/420 (31.0) | 208/999 (20.8) | 77/366 (21.0) | 188/1013 (18.6) | 161/348 (46.3) | 573/1211 (47.3) |
| Smoking, n (%) | | | | | | |
| Non smokers | 1072 (25.5) | 4275 (85.2) | 381 (28.0) | 1735 (86.0) | 438 (30.2) | 2041 (87.4) |
| Ex-smokers | 1036 (24.7) | 222 (4.4) | 409 (30.1) | 105 (5.2) | 393 (27.1) | 100 (4.3) |
| Regular smokers | 2091 (49.8) | 522 (10.4) | 571 (42.0) | 178 (8.6) | 620 (42.7) | 193 (8.3) |
| Alcohol intake, n (%) | | | | | | |
| Non drinkers | 829 (19.8) | 1822 (36.4) | 243 (17.9) | 624 (30.9) | 255 (17.67) | 705 (30.2) |
| Moderate drinkers | 2954 (70.5) | 3154 (62.9) | 979 (72.0) | 1380 (68.4) | 1046 (72.2) | 1612 (69.1) |
| Heavy drinkers | 409 (9.8) | 36 (0.7) | 137 (10.1) | 12 (0.6) | 148 (10.2) | 15 (0.6) |
| Education, n (%) | | | | | | |
| Primary | 470 (11.2) | 474 (9.4) | 64 (4.7) | 104 (5.2) | 82 (5.7) | 151 (6.5) |
| Secondary | 917 (21.6) | 1534 (30.6) | 275 (20.2) | 587 (29.1) | 323 (22.3) | 705 (30.2) |
| Vocational | 1473 (35.1) | 1685 (33.6) | 484 (35.6) | 720 (35.7) | 465 (32.0) | 754 (32.3) |
| University | 1339 (31.9) | 1326 (26.4) | 538 (39.6) | 607 (30.1) | 581 (40.0) | 724 (31.0) |
| Marital status, n (%) | | | | | | |
| Single | 507 (12.1) | 2031 (40.5) | 139 (10.2) | 764 (37.9) | 132 (9.1) | 855 (36.6) |
| Married /cohabiting | 3692 (87.9) | 2988 (59.5) | 1222 (89.8) | 1254 (62.1) | 1319 (90.9) | 1479 (63.4) |
| History of CVD, n (%) | 987 (23.5) | 1001 (19.9) | 309 (22.7) | 410 (19.9) | 266 (18.3) | 393 (16.8) |
| History of chronic diseases, n (%) | 1113 (26.5) | 1327 (26.4) | 361 (25.8) | 513 (25.4) | 302 (20.8) | 519 (22.2) |
| CVD death during follow-up ¯, n (%) | 374 (8.9) | 159 (3.3) | 69 (5.4) | 24 (1.2) | - | - |
| All causes death during follow-up **, n (%) | 839 (20.0) | 365 (7.3) | 141 (10.4) | 57 (2.8) | - | - |

Values are mean (sd) or n (%)

* LLT—lipid lowering treatment among those with history of dyslipidemia

** Mortality follow-up limited to 2011.

relatively stable in smokers while decreasing in never-smokers. LDL-C/HDL-C ratios were lower at baseline in smokers; trajectories were in opposition, increasing in ever-smokers and remained stable in never-smokers. These contrasting TG and LDL-C/HDL-C trajectories seem to have been further influenced by the age-cohort effect, with a more pronounced increase among smokers observed in subjects under 60 years of age and a decrease in those over 60 years of age (more details summarised within the S7–S12 Figs).

## Effects of lipid-lowering therapy

Fig 3 and S13 Fig show the differences in trajectories between subjects treated with lipid-lowering drugs and treatment-naïve subjects, adjusted for sex and centred age (58 years). As

**Table 2. Baseline total cholesterol, LDL-C, HDL-C and triglycerides (intercept) and change in LDL-C, HDL-C and triglycerides per year (slope) in the 5-yr cohorts (unadjusted).**

| | | Age range in W1 | TC | | | LDL-C | | |
|---|---|---|---|---|---|---|---|---|
| | | | coeff. | SE | p-value | coeff. | SE | p-value |
| **Intercept** | Estimate (mmol/l) | 45–49 (ref) | 5.94 | 0.031 | <0.001 | 3.76 | 0.028 | <0.001 |
| | Difference compared to reference group | 50–54 | 0.217 | 0.042 | <0.001 | 0.181 | 0.038 | <0.001 |
| | | 55–59 | 0.344 | 0.041 | <0.001 | 0.284 | 0.037 | <0.001 |
| | | 60–64 | 0.492 | 0.043 | <0.001 | 0.439 | 0.038 | <0.001 |
| | | 65–69 | 0.421 | 0.041 | <0.001 | 0.379 | 0.037 | <0.001 |
| **Slope** | Estimate (mmol/l/year) | 45–49 (ref) | -0.026 | 0.003 | <0.001 | -0.014 | 0.003 | <0.001 |
| | Difference compared to reference group | 50–54 | -0.025 | 0.005 | <0.001 | -0.021 | 0.004 | <0.001 |
| | | 55–59 | -0.047 | 0.005 | 0.001 | -0.041 | 0.004 | <0.001 |
| | | 60–64 | -0.065 | 0.005 | <0.001 | -0.053 | 0.004 | <0.001 |
| | | 65–69 | -0.069 | 0.006 | <0.001 | -0.059 | 0.005 | <0.001 |
| | | | HDL-C | | | TG | | |
| | | | coeff. | SE | p-value | coeff. | SE | p-value |
| **Intercept** | Estimate (mmol/l) | 45–49 (ref) | 1.54 | 0.010 | <0.001 | 1.40 | 0.020 | <0.001 |
| | Difference compared to reference group | 50–54 | -0.004 | 0.012 | 0.713 | 0.089 | 0.027 | 0.001 |
| | | 55–59 | -0.011 | 0.012 | 0.339 | 0.150 | 0.027 | <0.001 |
| | | 60–64 | -0.017 | 0.012 | 0.173 | 0.152 | 0.027 | <0.001 |
| | | 65–69 | -0.029 | 0.012 | 0.013 | 0.161 | 0.026 | <0.001 |
| **Slope** | Estimate (mmol/l/year) | 45–49 (ref) | -0.019 | 0.001 | <0.001 | 0.012 | 0.002 | <0.001 |
| | Difference compared to reference group | 50–54 | -0.002 | 0.002 | 0.178 | -0.006 | 0.003 | 0.053 |
| | | 55–59 | 0.001 | 0.002 | 0.761 | -0.015 | 0.003 | <0.001 |
| | | 60–64 | 0.002 | 0.002 | 0.368 | -0.028 | 0.003 | <0.001 |
| | | 65–69 | 0.003 | 0.002 | 0.101 | -0.030 | 0.003 | <0.001 |

expected, in subjects treated with lipid-lowering drugs, TC baseline values were lower; however, LDL-C, non-HDL-C and HDL-C values were similar, while TG values and LDL-C/HDL-C ratios were higher than in treatment-naïve subjects. The declines in all four main lipid categories over time (slope) in subjects given medication were much more pronounced than in untreated persons. For TG and LDL-C/HDL-C, trajectories were in opposite directions, decreasing in lipid-lowering-treated individuals and remaining stable or increasing in treatment-naïve subjects. More detailed comparison between the treated and untreated subjects within the different age subgroups are summarised within the S13–S18 Figs.

## Discussion

In this Russian population-based urban cohort aged 45–69 years old at baseline, we observed a reduction with aging in plasma levels of total, LDL- and HDL- cholesterol at individual level over 12 year follow up in all age groups. This decrease was more pronounced in subjects using cholesterol lowering medication and was more distinct for total cholesterol than to LDL-C. In contrast, plasma triglyceride levels increased with advancing age, with peak values observed in subjects in the age category of 55–59 years. Across all three examinations, the mean plasma lipid values were higher (lower in the case of HDL-cholesterol) than the values recommended by European cardiology societies [3]. As summarised previously, plasma concentrations of total and LDL-C in this population were 6.3 and 4.1 mmol/L, respectively [14].

This study has several limitations. First, we cannot exclude the possibility of differential cohort attrition due to increased CVD mortality risk in subjects with dyslipidaemia and/or

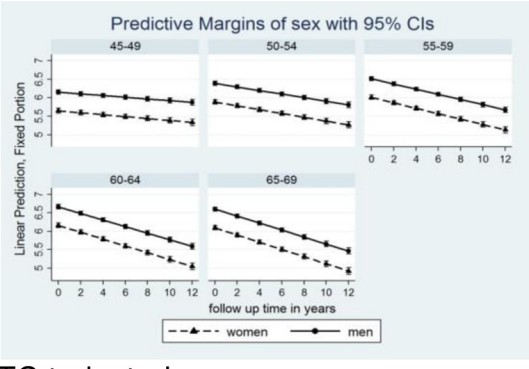

TC trajectories

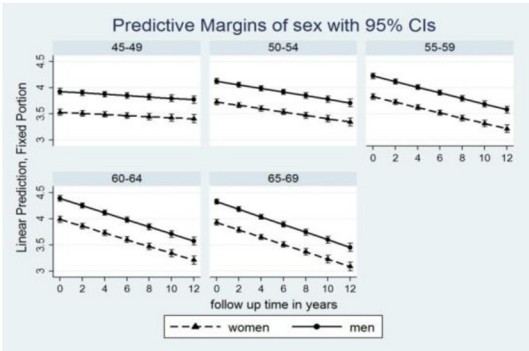

LDL-C trajectories

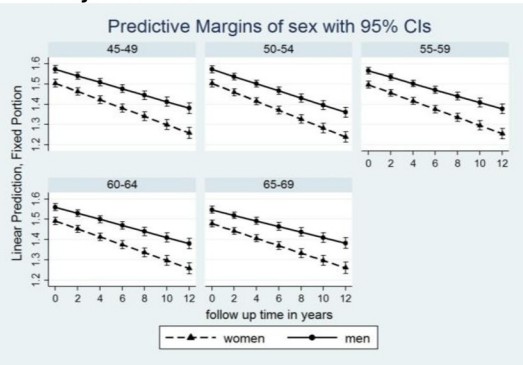

HDL-C trajectories

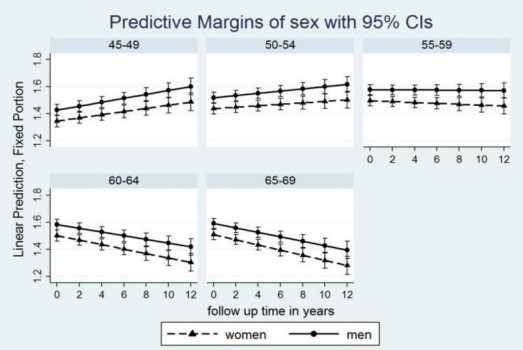

TG trajectories

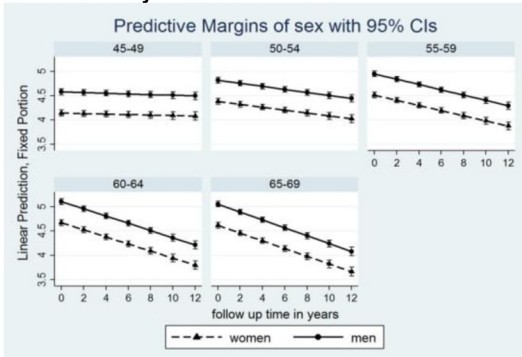

Non HDL-C trajectories

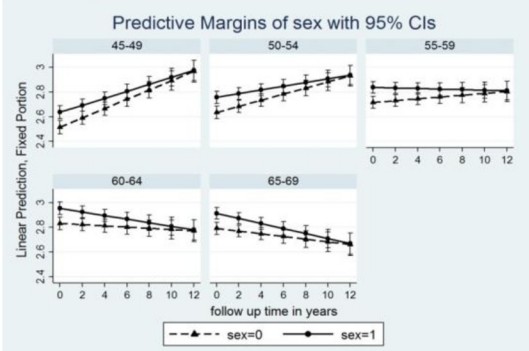

Ratio LDL-C/HDL-C trajectories

**Fig 1. Trajectories of lipids in men and women 5-yr cohorts over the 12 years of follow up.**

due to increased non-CVD mortality among those with extreme lipids decline. However, we adjusted the estimates for both total and CVD mortality. Importantly, our results were similar in full initial sample and two sensitivity analyses in 'complete case' sample (participants of all three waves) and after excluding persons who dropped out since baseline. This fact suggests that attrition did not affect the observed trajectories substantially.

Second, the baseline response rate was moderate (61%), which can affect the mean lipid levels, as non-participants in epidemiological studies usually have healthier risk factor profile. In addition, the study did not include young adults who may have otherwise exhibited different lipid trajectories.

**Table 3. Baseline total cholesterol, LDL-C, HDL-C and triglycerides (intercept) and change in LDL-C, HDL-C and triglycerides per year (slope) in the 5-yr cohorts (adjusted*).**

| | | Age range in W1 | TC | | | LDL-C | | |
|---|---|---|---|---|---|---|---|---|
| | | | coeff. | SE | p-value | coeff. | SE | p-value |
| **Intercept** | Estimate (mmol/l) | 45–49 (ref) | 5.64 | 0.033 | <0.001 | 3.54 | 0.030 | <0.001 |
| | Difference compared to reference group | 50–54 | 0.239 | 0.041 | <0.001 | 0.197 | 0.037 | <0.001 |
| | | 55–59 | 0.362 | 0.040 | <0.001 | 0.298 | 0.036 | <0.001 |
| | | 60–64 | 0.514 | 0.042 | <0.001 | 0.456 | 0.037 | <0.001 |
| | | 65–69 | 0.447 | 0.040 | <0.001 | 0.397 | 0.036 | <0.001 |
| **Slope** | Estimate (mmol/l/year) | 45–49 (ref) | -0.024 | 0.004 | <0.001 | -0.012 | 0.004 | 0.020 |
| | Difference compared to reference group | 50–54 | -0.026 | 0.005 | <0.001 | -0.021 | 0.004 | <0.001 |
| | | 55–59 | -0.047 | 0.005 | 0.001 | -0.041 | 0.004 | <0.001 |
| | | 60–64 | -0.066 | 0.005 | <0.001 | -0.054 | 0.004 | <0.001 |
| | | 65–69 | -0.071 | 0.005 | <0.001 | -0.060 | 0.004 | <0.001 |
| | | Age range in W1 | HDL-C | | | TG | | |
| | | | coeff. | SE | p-value | coeff. | SE | p-value |
| **Intercept** | Estimate (mmol/l) | 45–49 (ref) | 1.50 | 1.34 | 1.34 | 1.34 | 0.030 | <0.001 |
| | Difference compared to reference group | 50–54 | -0.011 | 0.091 | 0.091 | 0.091 | 0.037 | <0.001 |
| | | 55–59 | -0.007 | 0.150 | 0.150 | 0.150 | 0.036 | <0.001 |
| | | 60–64 | -0.012 | 0.154 | 0.154 | 0.154 | 0.037 | <0.001 |
| | | 65–69 | -0.025 | 0.163 | 0.163 | 0.163 | 0.036 | <0.001 |
| **Slope** | Estimate (mmol/l/year) | 45–49 (ref) | -0.020 | 0.013 | 0.013 | 0.013 | 0.004 | 0.020 |
| | Difference compared to reference group | 50–54 | -0.002 | -0.006 | -0.006 | -0.006 | 0.004 | <0.001 |
| | | 55–59 | 0.001 | -0.015 | -0.015 | -0.015 | 0.004 | <0.001 |
| | | 60–64 | 0.002 | -0.028 | -0.028 | -0.028 | 0.004 | <0.001 |
| | | 65–69 | 0.003 | -0.031 | -0.031 | -0.031 | 0.004 | <0.001 |

*Adjusted by sex and lipid-lowering treatment.

On the other hand, results were similar (concerning the direction and slope of the trajectories) when we repeated our assessment only in subjects whose serum lipids were known across all 3 waves.

We cannot fully exclude the impact of a range of potential confounding factors. However, we have adjusted the final models for the most common confounders (sex, age, lipid-lowering treatment, smoking, BMI, alcohol consumption, education, marital status, CVD/all-cause mortality); and also have stratified analysis by sex and by 5-year age groups. Few additional models were calculated controlling the lipid curves for basal fruits or vegetables intake (by tertiles) with centered value of total energy intake (10.6 MJ/day), and for leisure time physical activity (3 categories) (data not shown). These additional adjustments did not substantially change the trajectories direction. The specific consideration of nutrition or physical activity effect on lipid changes might be a subject of separate analysis (taking into account restrictions in data available) and it is not in the scope of present paper. Also we have not information on specific medications (except of class), but it is unlikely that separate drugs might affect the average lipid level beyond the general effect of lipid-lowering treatment.

Finally, this study only included participants from one large city, and these results may not be generalisable to the whole country. However, the trends of risk factors in Novosibirsk and rates of CVD mortality in the region are close to those seen the Russian Federation across the last 30 years (Rosstat), so Novosibirsk can be seen as a useful case study mirroring at some extent the national pattern.

**Table 4. Baseline total cholesterol, LDL-C, HDL-C and triglycerides (intercept) and change in LDL-C, HDL-C and triglycerides per year (slope) in the 5-yr cohorts (* multivariable adjusted).**

| | | Age range in W1 | TC | | | LDL-C | | |
|---|---|---|---|---|---|---|---|---|
| | | | coeff. | SE | p-value | coeff. | SE | p-value |
| **Intercept** | Estimate (mmol/l) | 45–49 (ref) | 5.35 | 0.058 | <0.001 | 3.37 | 0.052 | <0.001 |
| | Difference compared to reference group | 50–54 | 0.228 | 0.043 | <0.001 | 0.190 | 0.038 | <0.001 |
| | | 55–59 | 0.354 | 0.042 | <0.001 | 0.284 | 0.038 | <0.001 |
| | | 60–64 | 0.520 | 0.045 | <0.001 | 0.445 | 0.040 | <0.001 |
| | | 65–69 | 0.482 | 0.045 | <0.001 | 0.420 | 0.040 | <0.001 |
| **Slope** | Estimate (mmol/l/year) | 45–49 (ref) | -0.027 | 0.003 | <0.001 | -0.011 | 0.003 | 0.002 |
| | Difference compared to reference group | 50–54 | -0.025 | 0.005 | <0.001 | -0.021 | 0.004 | <0.001 |
| | | 55–59 | -0.045 | 0.005 | <0.001 | -0.039 | 0.004 | <0.001 |
| | | 60–64 | -0.064 | 0.005 | <0.001 | -0.052 | 0.004 | <0.001 |
| | | 65–69 | -0.068 | 0.007 | <0.001 | -0.057 | 0.005 | <0.001 |
| | | Age range in W1 | HDL-C | | | TG | | |
| | | | coeff. | SE | p-value | coeff. | SE | p-value |
| **Intercept** | Estimate (mmol/l) | 45–49 (ref) | 1.47 | 1.11 | 1.11 | 1.11 | 0.052 | <0.001 |
| | Difference compared to reference group | 50–54 | 0.020 | 0.059 | 0.059 | 0.059 | 0.038 | <0.001 |
| | | 55–59 | 0.021 | 0.098 | 0.098 | 0.098 | 0.038 | <0.001 |
| | | 60–64 | 0.014 | 0.131 | 0.131 | 0.131 | 0.040 | <0.001 |
| | | 65–69 | 0.017 | 0.094 | 0.094 | 0.094 | 0.040 | <0.001 |
| **Slope** | Estimate (mmol/l/year) | 45–49 (ref) | -0.018 | 0.008 | 0.008 | 0.008 | 0.003 | 0.002 |
| | Difference compared to reference group | 50–54 | -0.002 | -0.005 | -0.005 | -0.005 | 0.004 | <0.001 |
| | | 55–59 | -0.001 | -0.011 | -0.011 | -0.011 | 0.004 | <0.001 |
| | | 60–64 | -0.000 | -0.024 | -0.024 | -0.024 | 0.004 | <0.001 |
| | | 65–69 | 0.001 | -0.023 | -0.023 | -0.023 | 0.005 | <0.001 |

* Adjusted by sex, lipid-lowering treatment, smoking, BMI, alcohol intake, education, marital status, CVD mortality, all-cause mortality.

On the other hand, our study has also several important strengths. A single measurement of CVD risk factors is not sufficiently precise for estimating life-long disease risk [18, 19]. Baseline measurement of blood lipids as a CVD risk factor is prone to variations associated with factors such as age, lifestyle changes (mainly physical activity and dietary habits) and seasonal variability; e.g. in the case of total and LDL-C, fluctuations of more than 1 mmol/L between summer and winter seasons have been reported [20]. Thus, single measurements can be misleading, resulting in the misclassification of subjects into at-risk groups. Alternative models are now being used to improve risk estimation [21], such as variable categorisation of linear modelling. The analysis of trajectories from more than two individual measurements seems to offer more precise estimates for risk prediction [21, 22]. Supportive evidence is provided by recent Mendelian randomisation studies of lifetime risk of CVD [23, 24]. In UK Biobank study, Ference *et al.*, using score of genetic variants associated with lower LDL-C levels as instrument of randomization, reported strong association between lifelong genetic exposure to lower levels of LDL-C and lower cardiovascular risk [23]. Swedisch study [24] than suggest, that LDL-C is is related with myocardial infarction, but not with stroke or heart failure.

Importantly, this large cohort study models long-term trajectories of lipids in correlation with advancing age. To the best of our knowledge, no similar study has focused on lipid trajectories within the Russian population (a group on which there is a significant lack of epidemiological data) or anywhere in Central or Eastern Europe. Thus, our study goes some way to redressing this deficit in knowledge of plasma lipids in this large region. In addition to

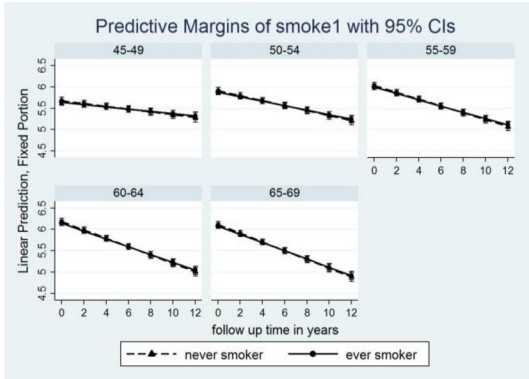

TC trajectories

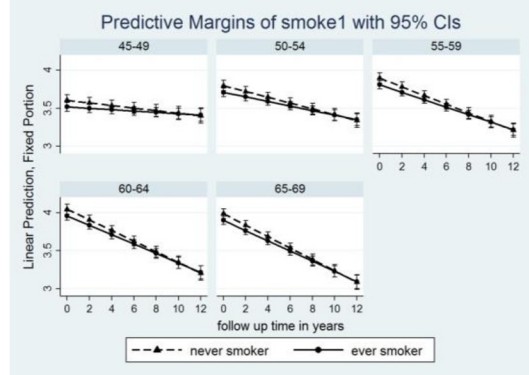

LDL-C trajectories

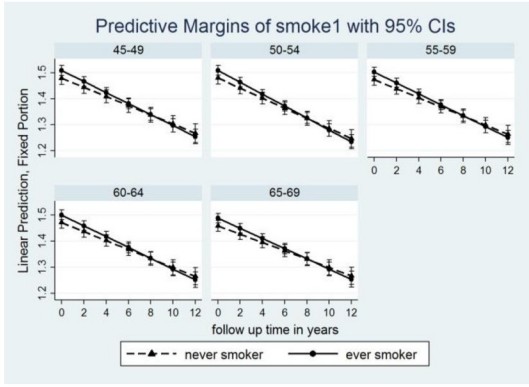

HDL-C trajectories

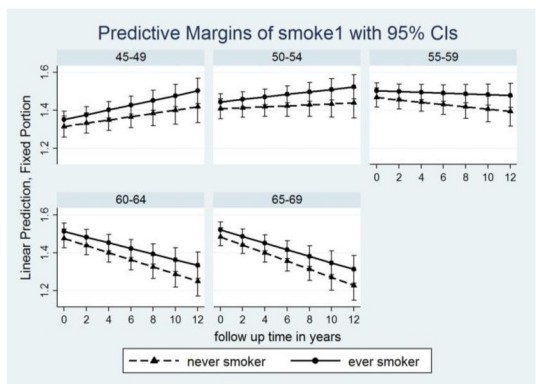

TG trajectories

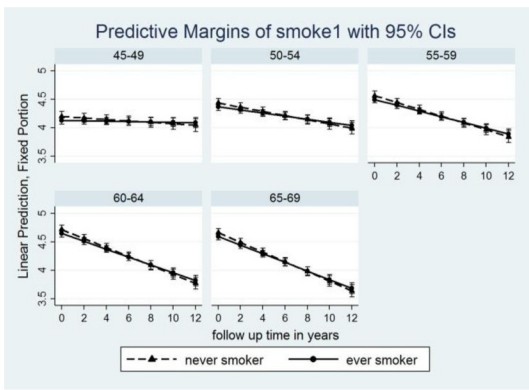

Non HDL-C trajectories

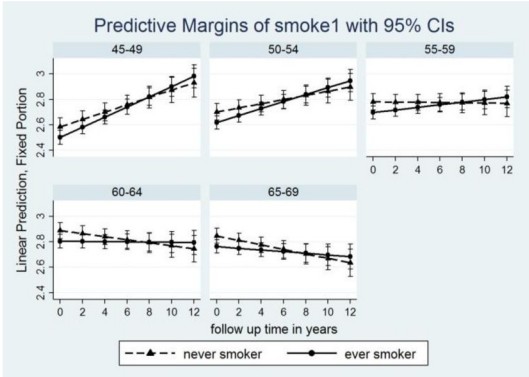

Ratio LDL-C/HDL-C trajectories

**Fig 2. Trajectories of lipids in men and women 5-yr cohorts over the 12 years of follow up by smoking.**

measuring traditional lipids, we also examined non-HDL-C, currently defined as the most relevant atherogenic lipid marker [25].

Together with smoking, diabetes, hypertension and obesity, plasma lipid values (especially total and LDL-cholesterol values) have long been considered as a key modifiable risk factor for atherosclerosis and cardiovascular disease. As observed in the WHO MONICA and post-MONICA studies, plasma lipid values have declined substantially over recent decades due to

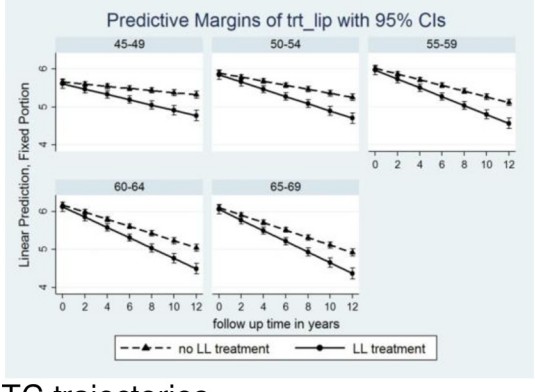

TC trajectories

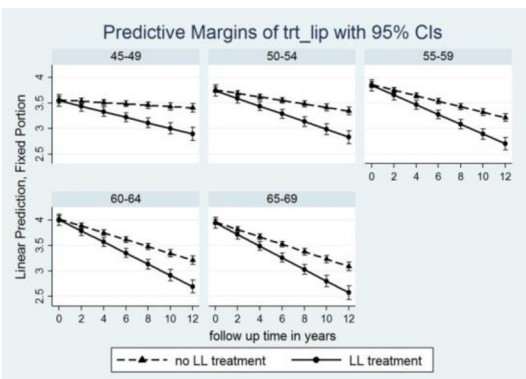

LDL-C trajectories

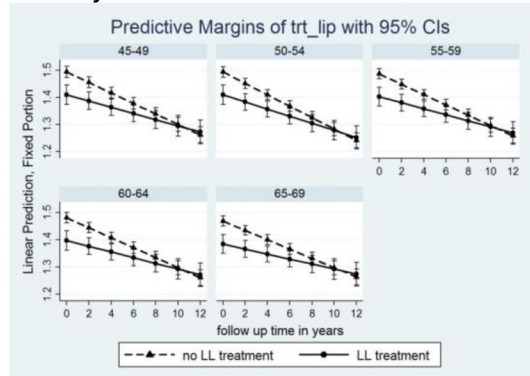

HDL-C trajectories

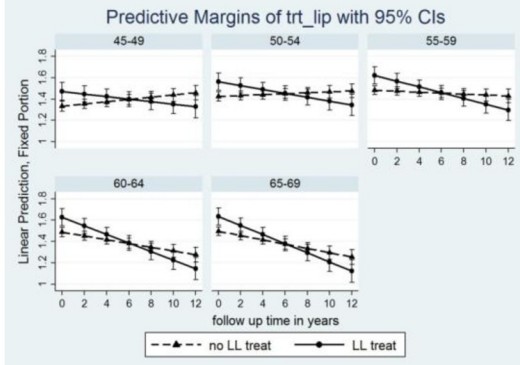

TG trajectories

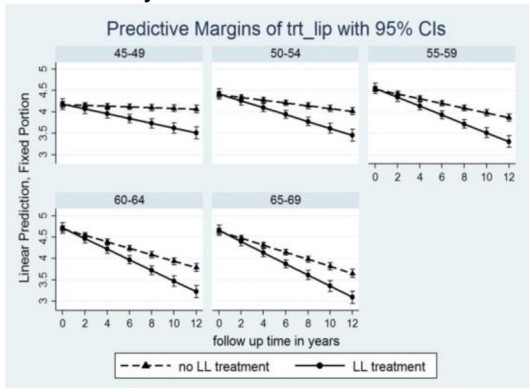

Non HDL-C trajectories

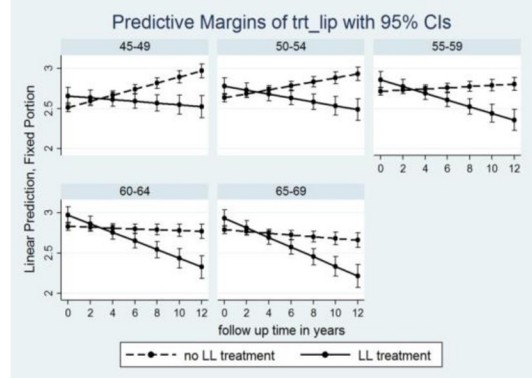

Ratio LDL-C/HDL-C

**Fig 3. Trajectories of lipids in men and women 5-yr cohorts over the 12 years of follow up by lipid lowering treatment.**

the wide use of statins and the adoption of positive lifestyle changes [26, 27]. In turn, this trend has led to changes in cardiovascular disease and mortality prediction [28–31].

The significant decline in plasma cholesterol values across age groups is consistent with the smaller effect of plasma cholesterol on CVD risk at older ages compared to younger groups. The reasons for the decline in levels of major lipids with age in the Russian population are not easily explained. As mentioned above, the decline appears to be largely treatment-independent. Lifestyle data on dietary habits and physical activity may provide some clues. For instance, the Russian diet is typically higher in energy and lower in vegetable and fruit intake compared to the Western European diet [32, 33], while physical activity in the Russian

population is relatively comparable to both Western and Eastern European populations [Malyutina et al., unpublished data]. Unfortunately, available data about dietary habits and/or general physical activities of examined subjects were not known in sufficient details to perform adequate analyses of effects.

While cross-sectional decline in lipids with age in elderly has been widely reported in the literature [6–8], analyses of long-term individual-level trajectories charting the development of lipid parameters remain scarce. One of the first analyses was reported in the Framingham Heart Study [34]. Similar trajectories have been reported by Swiger *et al.* [35], who describe an age-related decline in LDL-C in subjects over 60 years of age. Our results are in agreement with these studies. According to the findings of National Health and Nutrition Examination Surveys [36], age-related decline may contribute to the decreased cholesterol levels seen in the population over recent decades.

The individual-level decline in blood lipid values in older subjects observed in our longitudinal study reflects similar trends documented in cross-sectional studies [36, 37]. Our findings are consistent with the ARIC Study, which followed up a sample population comprising 9,328 subjects of European ancestry over 9 years. The authors observed an overall decrease in TC and LDL-C levels, a relative stabilisation in HDL-C levels, and a consistent rise in plasma TG values [8]. Similar patterns were very recently reported in 35-year follow-up of Framingham Offspring participants using the group modelling technique [11].

In contrast to TC and LDL-C, trajectories of serum TG levels in our cohorts varied substantially between age groups. Generally, we observed an increase in serum TG values in younger age groups, a plateau in the 55-59-years age category, followed by a decline in subjects over 60 years of age. This is consistent with the recent suggestions that serum TG and TG-rich remnant particles and non-HDL-C might be more powerful for predicting CVD and total mortality than TC or LDL-C [37]. Based on our previous observations of another geographically identical population (the Novosibirsk MONICA cohort), high TG values significantly increased the 10-year risk of CVD outcomes in men and, especially, women [38].

We have observed an effect of smoking status on lipid trajectories. While TC values were similar in smokers and never smokers, with both groups showing similar decreasing slopes, HDL-C was higher at baseline among smokers but decreased faster, so that at the end of the follow up HDL-C became higher among never-smokers. TG values where higher in ever-smokers in comparison with never-smokers, and the observed declining trajectory was steeper in never-smokers.

There were some differences between men and women in lipid trajectories. In general, the trajectories in males were steeper than in females. Revealingly, in the youngest age category, the trajectories of LDL-C/HDL-C ratios were significantly higher in males than in females, but no differences between males and females were observed for the highest age category.

An important finding of this study is the parallel decline of serum total and LDL-C in both treatment-naïve and statin-treated subjects. Although the declining slope was, as expected, slightly steeper in treated subjects, it does not seem that the decline in total and LDL cholesterol in the whole population can be attributed exclusively to lipid-lowering drugs. The use of statins, the major lipid–lowering drug group, increased from 17.9% to 19% in entire population (from 23.8% to 47.1% among those aware of dyslipidaemia). Interestingly, there were no subjects on ezetimibe or PCSK9 inhibitors, as on the most powerful lipid lowering drug known so far, included. The treatment with other lipid-lowering drugs has been negligible (around 1%).

The steeper decline trajectories among subjects treated with lipid lowering drugs compared to those not treated in our sample are consistent with the patterns recently shown by Duncan *et al.* [10] in Framingham Offspring Study. In the groups composed from individuals with

initially elevated level of TC, LDL-C and non-HDL-C which were decreasing with age, the proportion of lipid lowering treatment was the highest.

## Conclusions

In this study in older Russians using the multilevel modelling of longitudinal lipid trajectories, we observed substantial declines in total cholesterol and LDL-C over time and with age. The pattern of declining concentrations does not seem to be attributable solely to lipid-lowering treatment and/or smoking behaviour. On the whole, elderly people seems to be at lower risk of suffering from dyslipidaemia, and gender differences become smaller with increasing age. Longitudinal trajectories of blood lipids from middle to the elderly age may improve prediction of CVD risk beyond using single occasion measurements and may improve the quantification of lifetime exposure to blood lipid concentrations.

## Supporting information

**S1 Table. Complete cases of all three waves (N = 2009).** Baseline total cholesterol, LDL-C, HDL-C and triglycerides (intercept) and change in LDL-C, HDL-C and triglycerides per year (slope) in the 5-yr cohorts (unadjusted).
(DOCX)

**S2 Table. Complete cases of all three waves (N = 2009).** Baseline total cholesterol, LDL-C, HDL-C and triglycerides (intercept) and change in LDL-C, HDL-C and triglycerides per year (slope) in the 5-yr cohorts. Adjusted by sex, lipid-lowering treatment, smoking, BMI, alcohol, education, marital status, CVD mortality, all causes mortality.
(DOCX)

**S3 Table. Selected sample after exclusion of subjects dropped out after baseline examination and did not die during follow-up (N = 7,606).** Baseline total cholesterol, LDL-C, HDL-C and triglycerides (intercept) and change in LDL-C, HDL-C and triglycerides per year (slope) in the 5-yr cohorts (unadjusted).
(DOCX)

**S4 Table. Selected sample after exclusion of dropped out after baseline examination and did not die during follow-up (N = 7,606).** Baseline total cholesterol, LDL-C, HDL-C and triglycerides (intercept) and change in LDL-C, HDL-C and triglycerides per year (slope) in the 5-yr cohorts. Adjusted by sex, lipid-lowering treatment, smoking, BMI, alcohol, education, marital status, CVD mortality, all causes mortalit.
(DOCX)

**S1 Fig. TC trajectories in men and women over the 12 years of follow up.**
(DOCX)

**S2 Fig. LDL-C trajectories in men and women over the 12 years of follow up.**
(DOCX)

**S3 Fig. HDL-C trajectories in men and women over the 12 years of follow up.**
(DOCX)

**S4 Fig. TG trajectories in men and women over the 12 years of follow up.**
(DOCX)

**S5 Fig. Non HDL-C trajectories in men and women over the 12 years of follow up.**
(DOCX)

**S6 Fig. Ratio LDL/HDL trajectories in men and women over the 12 years of follow up.**
(DOCX)

**S7 Fig. TC trajectories in men and women over the 12 years of follow up by smoking.**
(DOCX)

**S8 Fig. LDL-C trajectories in men and women over the 12 years of follow up by smoking.**
(DOCX)

**S9 Fig. HDL-C trajectories in men and women over the 12 years of follow up by smoking.**
(DOCX)

**S10 Fig. TG trajectories in men and women over the 12 years of follow up by smoking.**
(DOCX)

**S11 Fig. NonHDL-C trajectories in men and women over the 12 years of follow up by smoking.**
(DOCX)

**S12 Fig. Ratio LDL/HDL trajectories in men and women over the 12 years of follow up by smoking.**
(DOCX)

**S13 Fig. TC trajectories in men and women over the 12 years of follow up by lipid lowering treatment.**
(DOCX)

**S14 Fig. LDL-C trajectories in men and women over the 12 years of follow up by lipid lowering treatment.**
(DOCX)

**S15 Fig. HDL-C trajectories in men and women over the 12 years of follow up by lipid lowering treatment.**
(DOCX)

**S16 Fig. TG trajectories in men and women over the 12 years of follow up by lipid lowering treatment.**
(DOCX)

**S17 Fig. nonHDL-C trajectories in men and women over the 12 years of follow up by lipid lowering treatment.**
(DOCX)

**S18 Fig. Ratio LDL/HDL trajectories in men and women over the 12 years of follow up by lipid lowering treatment.**
(DOCX)

## Author Contributions

**Conceptualization:** Jaroslav A. Hubacek, Michael V. Holmes, Martin Bobak, Sofia Malyutina.

**Data curation:** Yuri Nikitin, Yulia Ragino, Ekaterina Stakhneva, Andrey Ryabikov, Eugeny Verevkin.

**Formal analysis:** Hynek Pikhart, Denes Stefler, Eugeny Verevkin, Sofia Malyutina.

**Funding acquisition:** Martin Bobak, Sofia Malyutina.

**Investigation:** Jaroslav A. Hubacek, Yuri Nikitin, Yulia Ragino, Ekaterina Stakhneva, Martin Bobak, Sofia Malyutina.

**Methodology:** Jaroslav A. Hubacek, Hynek Pikhart, Anne Peasey, Michael V. Holmes, Martin Bobak, Sofia Malyutina.

**Project administration:** Andrey Ryabikov, Eugeny Verevkin, Sofia Malyutina.

**Resources:** Martin Bobak, Sofia Malyutina.

**Supervision:** Jaroslav A. Hubacek, Hynek Pikhart, Anne Peasey, Michael V. Holmes, Denes Stefler, Martin Bobak.

**Visualization:** Sofia Malyutina.

**Writing – original draft:** Jaroslav A. Hubacek.

**Writing – review & editing:** Yuri Nikitin, Yulia Ragino, Ekaterina Stakhneva, Hynek Pikhart, Anne Peasey, Michael V. Holmes, Denes Stefler, Andrey Ryabikov, Eugeny Verevkin, Martin Bobak, Sofia Malyutina.

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
