## [Decision Letter · Decision Letter 0]

23 Aug 2021

PONE-D-21-21255

Longitudinal trajectories of blood lipid levels in an ageing population sample of Russian Western-Siberian urban population

PLOS ONE

Dear Dr. Hubacek,

Thank you for submitting your manuscript to PLOS ONE. After careful consideration, we feel that it has merit but does not fully meet PLOS ONE’s publication criteria as it currently stands. Therefore, we invite you to submit a revised version of the manuscript that addresses the points raised during the review process.

Please consider all the reviewer's comments and address properly all of them.

We look forward to receiving your revised manuscript.

Kind regards,

Paolo Magni

Academic Editor

PLOS ONE

“This study was supported by the Russian Scientific Foundation (20-15-00371,14-45-00030) and the Russian Academy of Science, State target (АААА-А17-117112850280-2); the HAPIEE study was funded by the Welcome Trust (WT064947, WT081081), the US National Institute of Aging (1RO1AG23522) and the MacArthur Foundation Initiative on Social Upheaval and Health; JAH is supported by the Ministry of Health of the Czech Republic under the Conceptual Development of Research Organisations project (Institute for Clinical and Experimental Medicine – IKEM, IN 00023001). MVH works in a unit that receives funding from the UK Medical Research Council and is supported by a British Heart Foundation Intermediate Clinical Research Fellowship (FS/18/23/33512) and the National Institute for Health Research Oxford Biomedical Research Centre.”

Funding information should not appear in the Acknowledgments section or other areas of your manuscript. We will only publish funding information present in the Funding Statement section of the online submission form.

“This study was supported by the Russian Scientific Foundation (20-15-00371,14-45-00030) and the Russian Academy of Science, State target (АААА-А17-117112850280-2); the HAPIEE study was funded by the Welcome Trust (WT064947, WT081081), the US National Institute of Aging (1RO1AG23522) and the MacArthur Foundation Initiative on Social Upheaval and Health; JAH is supported by the Ministry of Health of the Czech Republic under the Conceptual Development of Research Organisations project (Institute for Clinical and Experimental Medicine – IKEM, IN 00023001). MVH works in a unit that receives funding from the UK Medical Research Council and is supported by a British Heart Foundation Intermediate Clinical Research Fellowship (FS/18/23/33512) and the National Institute for Health Research Oxford Biomedical Research Centre. Sponsors play no role in the study design, data collection and analysis, decision to publish, or preparation of the manuscript.”

5. We note that you have referenced (Malyutina et al.,) which has currently not yet been accepted for publication. Please remove this from your References and amend this to state in the body of your manuscript: (ie “Malyutina et al., unpublished data”) as detailed online in our guide for authors

Reviewers' comments:

Reviewer's Responses to Questions

**Comments to the Author**

1. Is the manuscript technically sound, and do the data support the conclusions?

Reviewer #1: Yes

2. Has the statistical analysis been performed appropriately and rigorously? 

Reviewer #1: Yes

3. Have the authors made all data underlying the findings in their manuscript fully available?

Reviewer #1: Yes

4. Is the manuscript presented in an intelligible fashion and written in standard English?

Reviewer #1: Yes

5. Review Comments to the Author

Reviewer #1: The manuscript is written in clear English. For achieving the aim of the study (evaluation of changes in blood lipid concentrations associated with age in a Russian West-Siberian Caucasian population) adequate statistical methods for calculations and sample size are used. The idea of the manuscript is structured and logical based on the purpose declared. However, it could be considered as a one part of a project aiming to reclassify current cardiovascular risk scores as

- it does not take into account a range of other confounding factors (diet particularities, other medications that may affect lipid blood levels, physical activity etc) that may affect the curve of the blood lipid changes,

- the mendelian randomization data about the life-time cardiovascular risk impact has to be taken into account,

- the results presented in the manuscript have to be supported by the data about the logic association of the blood lipid changes curve in different age groups with the prevalence of cardiovascular events in these groups.

6. PLOS authors have the option to publish the peer review history of their article (what does this mean?). If published, this will include your full peer review and any attached files.

Reviewer #1: No

---

## [Author Response · Author response to Decision Letter 0]

3 Nov 2021

Comments to the Author

1/ - it does not take into account a range of other confounding factors (diet particularities, other medications that may affect lipid blood levels, physical activity etc) that may affect the curve of the blood lipid changes

RE: We agree that in addition to lipid lowering treatment, age, sex and smoking, there are other factors that could influence plasma lipid levels in study participants. Concerning the lipid lowering treatment, as mentioned in the discussion, our subjects were almost exclusively treated by statins (no subjects on PCSK9 inhibitors or ezetimibe were included; see now added information at lines 460 - 463). In terms of other medications – unfortunately in the HAPIEE study we have no data available on specific medication of participants apart from lipid-lowering drugs, antihypertensive drugs or sugar lowering drugs (on dosage or regiment of intake). We have described this limitation of our data in the Discussion section of the revised manuscript.

In additional sensitivity analyses, we have further adjusted the models for fruit or vegetables intakes (by tertiles); total energy intake (centered at 10.6 MJ/day); and leisure time physical activity (less than 3 MET-hours/day; 3-10 MET hours/day; more than 10 MET hours/day). These adjustments did not materially change the results, including the direction of trajectories. We have added the description of additional adjustments to the revised paper (lines 353 - 366). Unfortunately, available data about dietary habits and/or general physical activities of examined subjects were not known in sufficient details at all three examinations to perform adequate analyses of effects. This is now listed between the study weaknesses (lines 414-416).

2/ - the mendelian randomization data about the life-time cardiovascular risk impact has to be taken into account

 RE: Thank you for this comment raising the issue of Mendelian randomisation that we have now incorporated to the Discussion. We agree that Mendelian randomisation data could be of interest for the estimation of life exposure to the distinct risk factors and their life-time impact on CVD risk. However, such analysis has not been performed in our study, mainly because of unavailable genotyping data due to the shortage of funding, and this approach is out of the scope of present paper.

Nevertheless, it is merit to refer the recent papers which used the randomization by genetic score based on polymorphisms associated with LDL-C (known from GWAS), and reported the association between life-long LDL-C lowering and reduced CVD risk [Ference et al., 2019; Lind et al., 2021, ref list under 23 and 24]. In revised manuscript we have widened Discussion taking into account these findings (lines 384 - 389).

3/ - the results presented in the manuscript have to be supported by the data about the logic association of the blood lipid changes curve in different age groups with the prevalence of cardiovascular events in these groups.

RE: Thank you for this comment. In our analysis, we have taken into account CVD of participants by adjusting for CVD mortality rather than prevalence.

As mentioned in the Methods (lines – 161 - 162), Results (Table 4 and lines 242 - 247) and Discussion (lines – 337 - 345) sections, both total and cardiovascular mortality has been taken into account during the calculations – we have adjusted our results on these parameters. Two sensitivity analyses also suggest that the effect of CVD mortality on lipid trajectories is small and did not affect the observed trajectories substantially (see lines 340 - 345). Unadjusted and fully adjusted results are presented in Tables 2 and 4. In Tables 2, 3, 4 and wide list of supplementary figures, data are also presented for different age groups (as suggested by reviewer).

---

## [Editor Report · Decision Letter 1]

5 Nov 2021

Longitudinal trajectories of blood lipid levels in an ageing population sample of Russian Western-Siberian urban population

PONE-D-21-21255R1

Dear Dr. Hubacek,

We’re pleased to inform you that your manuscript has been judged scientifically suitable for publication and will be formally accepted for publication once it meets all outstanding technical requirements.

Kind regards,

Paolo Magni

Academic Editor

PLOS ONE

---

## [Editor Report · Acceptance letter]

17 Nov 2021

PONE-D-21-21255R1 

Longitudinal trajectories of blood lipid levels in an ageing population sample of Russian Western-Siberian urban population 

Dear Dr. Hubacek:

I'm pleased to inform you that your manuscript has been deemed suitable for publication in PLOS ONE. Congratulations! Your manuscript is now with our production department. 

Kind regards, 

on behalf of

Prof. Paolo Magni 

Academic Editor

PLOS ONE